# ANGPTL2 expression in the intestinal stem cell niche controls epithelial regeneration and homeostasis

Haruki Horiguchi[1,2,*] (iD), Motoyoshi Endo[1,**] (iD), Kohki Kawane[3], Tsuyoshi Kadomatsu[1], Kazutoyo Terada[1], Jun Morinaga[1], Kimi Araki[2], Keishi Miyata[1] & Yuichi Oike[1,***] (iD)

## Abstract

The intestinal epithelium continually self-renews and can rapidly regenerate after damage. Dysregulation of intestinal epithelial homeostasis leads to severe inflammatory bowel disease. Additionally, aberrant signaling by the secreted protein angiopoietin-like protein 2 (ANGPTL2) causes chronic inflammation in a variety of diseases. However, little is known about the physiologic role of ANGPTL2 in normal tissue homeostasis and during wound repair following injury. Here, we assessed ANGPTL2 function in intestinal physiology and disease *in vivo*. Although intestinal development proceeded normally in *Angptl2*-deficient mice, expression levels of the intestinal stem cell (ISC) marker gene *Lgr5* decreased, which was associated with decreased transcriptional activity of β-catenin in *Angptl2*-deficient mice. Epithelial regeneration after injury was significantly impaired in *Angptl2*-deficient relative to wild-type mice. ANGPTL2 was expressed and functioned within the mesenchymal compartment cells known as intestinal subepithelial myofibroblasts (ISEMFs). ANGPTL2 derived from ISEMFs maintained the intestinal stem cell niche by modulating levels of competing signaling between bone morphogenetic protein (BMP) and β-catenin. These results support the importance of ANGPTL2 in the stem cell niche in regulating stemness and epithelial wound healing in the intestine.

**Keywords** ANGPTL2; BMP; homeostasis; ISEMF; regeneration
**Subject Categories** Signal Transduction; Stem Cells
The EMBO Journal (2017) 36: 409–424

See also: **SM van Neerven & L Vermeulen** (February 2017)

## Introduction

External and internal stresses cause structural and functional tissue damage in various organs. Such damage is repaired by tissue remodeling mechanisms governed by signaling between parenchymal and stromal cells via cell–cell contact or humoral factors (Medzhitov, 2008). The intestine presents a unique model in which to study mammalian tissue homeostasis. The intestinal lumen is susceptible to external and internal stresses, and the intestinal epithelium is characterized by rapid and continuous renewal throughout an animal's life to re-establish the epithelial barrier after mucosal injury (Heath, 1996). However, continuous unresolved inflammation and pathological irreversible tissue injury due to breakdown in tissue homeostasis leads to severe inflammatory bowel disease (IBD), leading to intestinal tissue damage and some forms of cancer. Thus, a better understanding of cellular and molecular mechanisms underlying tissue homeostasis could provide insight into the etiology of IBD.

Intestinal homeostasis is regulated by proliferation and differentiation of cycling intestinal stem cells (ISCs), which express the surface markers LGR5, ASCL2, and OLFM4 (Barker *et al*, 2007; Barker, 2014). LGR5-positive stem cells actively proliferate and differentiate into all cell types seen in the intestine, regulated in part by the surrounding microenvironment, known as the stem cell niche (Yen & Wright, 2006; Walker *et al*, 2009). Intestinal subepithelial myofibroblasts (ISEMFs) located immediately subjacent to ISCs provide important paracrine regulatory signals during normal physiologic turnover and in the context of wound repair (Otte *et al*, 2003; Powell *et al*, 2011; Chivukula *et al*, 2014). Several signaling pathways, including Wnt, bone morphogenetic protein (BMP), Notch, and Hedgehog, reportedly regulate the fate of ISCs (Medema & Vermeulen, 2011; Sato *et al*, 2011b), and intestinal homeostasis is regulated by opposing gradients of BMP and Wnt/β-catenin signaling. Stem cell expansion is greatest at the crypt base, where Wnt/β-catenin signaling is highest, and transit amplifying (TA) cells undergo proliferation (Reya & Clevers, 2005). By contrast, BMP signaling, which inhibits proliferation, is highest at the luminal surface (Wakefield & Hill, 2013) and inhibited at the crypt base by the BMP antagonists Noggin, GREM1, and GREM2 (He *et al*, 2004; Kosinski *et al*, 2007). In humans, perturbed β-catenin/BMP signaling is associated with juvenile polyposis syndrome (JPS), familial adenomatous polyposis (FAP), and colorectal cancer (van Es *et al*, 2001; Howe *et al*, 2001; Waite & Eng, 2003). However, how

1   Department of Molecular Genetics, Graduate School of Medical sciences, Kumamoto University, Chuo-ku, Kumamoto, Japan
2   Institute of Resource Development and Analysis, Kumamoto University, Chuo-ku, Kumamoto, Japan
3   Faculty of Life Sciences, Kyoto Sangyo University, Kita-ku, Kyoto, Japan
    *Corresponding author. Tel: +81 96 373 5140; Fax: +81 96 373 5145; E-mail: horiguti@kumamoto-u.ac.jp
    **Corresponding author. Tel: +81 96 373 5140; Fax: +81 96 373 5145; E-mail: enmoto@gpo.kumamoto-u.ac.jp
    ***Corresponding author. Tel: +81 96 373 5140; Fax: +81 96 373 5145; E-mail: oike@gpo.kumamoto-u.ac.jp

β-catenin/BMP integration is regulated at the cellular or molecular level remains unclear.

Recently, we reported that aberrant angiopoietin-like protein 2 (ANGPTL2) signaling causes chronic inflammatory diseases, such as obesity, metabolic disease, type 2 diabetes, atherosclerotic disease, and possibly some cancers (Oike & Tabata, 2009; Tabata *et al*, 2009; Aoi *et al*, 2011; Endo *et al*, 2012; Horio *et al*, 2014). In intestine, tumor cell-derived ANGPTL2 renders colorectal cancer cells resistant to chemotherapy via anti-apoptotic signaling (Horiguchi *et al*, 2014). Nevertheless, ANGPTL2 function in intestine in physiologic conditions has not been investigated.

Here, we assessed ANGPTL2 function in intestinal physiology and disease. In contrast to its dispensable function in normal intestinal development, ANGPTL2 depletion impaired intestinal regeneration following dextran sulfate sodium (DSS) and irradiation-induced epithelial injury. We report that ANGPTL2 is expressed by mice ISEMFs and that ISEMF-derived ANGPTL2 inhibits *Bmp* mRNA induction in an autocrine manner, maintaining ISC stemness by BMP/β-catenin signaling in intestinal epithelial cells (IECs). These findings show that ANGPTL2 expression in the ISC niche is important to regulate intestinal epithelial regeneration and homeostasis.

# Results

### β-Catenin signaling decreases in the *Angptl2*$^{-/-}$ mouse colon

To assess intestinal ANGPTL2 function, we initially undertook histological examination of intestine in adult wild-type and *Angptl2*-deficient (*Angptl2*$^{-/-}$) mice. The architecture of intestinal crypt as well as cell proliferation was comparable in both genotypes based on hematoxylin and eosin (H&E) staining (Figs 1A and EV1A) and PAS staining (goblet cell) (Fig 1B), immunochemistry (IHC) for Ki67 (a proliferation marker) (Fig 1C), and immunofluorescence (IF) for phosphorylated histone H3 (pH3, a mitotic marker) (Fig 1D). In addition, bromodeoxyuridine (BrdU) pulse-chase experiments showed that BrdU uptake and epithelial turnover rates were equivalent in colon cells of both genotypes (Fig EV1B) and in small intestine (Fig EV1C), suggesting overall that ANGPTL2 is not required for normal intestinal development and does not appreciably regulate baseline epithelial cell turnover in intestine. By contrast, crypts from *Angptl2*$^{-/-}$ mice showed lower levels of transcripts encoding ISC markers, namely *Lgr5* and to a lesser extent *Ascl2*, despite the fact that expression of the cell proliferation markers *Myc* and *Ccnd1* was comparable (Fig 1E).

Wnt/ β-catenin signaling is indispensable for stem cell expansion and crypt formation (Pinto *et al*, 2003; Fevr *et al*, 2007; Hirata *et al*, 2013). Because crypts isolated from *Angptl2*$^{-/-}$ mice showed lower levels of ISC marker genes, we examined β-catenin expression in wild-type and *Angptl2*$^{-/-}$ mouse crypts. Western blot analysis revealed that total β-catenin protein in crypts decreased in *Angptl2*$^{-/-}$ mice (Fig 1F). Phosphorylated β-catenin (P-β-catenin) is unstable and undergoes degradation (Yost *et al*, 1996), while non-phosphorylated (active) β-catenin protein has transcriptional activity (Sakanaka, 2002). Levels of P-β-catenin increased and those of active β-catenin decreased in *Angptl2*$^{-/-}$ mouse crypts (Fig 1F), and lower levels of active β-catenin were detected in the nucleus of cells in crypts in *Angptl2*$^{-/-}$ relative to wild-type mice (Fig 1G),

suggesting that ANGPTL2 functions in β-catenin signaling associated with intestinal regeneration.

### ANGPTL2 is important for intestinal regeneration

Oral administration of DSS reportedly induces colitis in mouse models due to loss of intestinal barrier function and formation of colon lesions (Okayasu *et al*, 1990). In these paradigms, intestinal regeneration marked by rapid crypt hyperplasia, enlargement, and fission is initiated after DSS withdrawal from drinking water (Okayasu *et al*, 1990; Clapper *et al*, 2007). To assess a potential function for ANGPTL2 in intestinal regeneration, we administered DSS in drinking water to wild-type and *Angptl2*$^{-/-}$ mice for 6 days. We then switched animals to normal water without DSS for an additional 6 days and monitored potential pathophysiologic changes in intestine. We observed upregulated *Angptl2* mRNA in wild-type mice colon following DSS treatment (Fig EV2A), suggesting a potential function in intestinal regeneration. By day 12, *Angptl2*$^{-/-}$ mice had shown increased mortality relative to wild-type mice (Fig 2A). *Angptl2*$^{-/-}$ mice also displayed rapid onset of weight loss at day 1 (Fig EV2B) and rectal bleeding and diarrhea associated with an increase in the disease activity index compared to wild-type mice (Fig EV2C). *Angptl2*$^{-/-}$ mice also showed marked shortening of the colon compared to wild-type mice at day 12 (Fig 2B and C). By day 12, when mice had received normal drinking water for 6 days, wild-type mice showed typical regenerative responses, characterized by the appearance of enlarged, hyperplastic crypts (Figs 2D and EV2D), while *Angptl2*$^{-/-}$ mice showed dramatic reduced numbers of crypts (Fig 2D). We then quantified crypt phenotypes by counting viable crypts, which we defined as those containing Ki67-positive cells within a crypt-like structure (Figs 2E and EV2E) or pH3-positive cells (Figs 2F and EV2F). We observed that *Angptl2*$^{-/-}$ mice showed reduced numbers of viable crypts and pH3-positive cells at day 12 (Fig 2G and H). In addition, lower levels of active β-catenin were detected in the nucleus of cells in crypts in *Angptl2*$^{-/-}$ mice relative to wild-type mice after DSS treatment for 6 days on and 6 days off (Fig 2I). In fact, induction of genes functioning in proliferation, such as *Myc* and *Ccnd1* mRNAs, was reduced in crypts from *Angptl2*$^{-/-}$ mice after DSS treatment for 6 days on and 6 days off (Fig 2J). In addition, levels of *Lgr5* and *Ascl2* transcripts, which are ISC markers regulated by β-catenin, decreased in crypts from *Angptl2*$^{-/-}$ compared to wild-type mice at day 12 (Fig 2J).

We next assessed intestinal regeneration using a different injury model that employed whole-body irradiation. For this procedure, mice were irradiated one time (12 Gy) and then assessed phenotypically at various time points up to 5 days thereafter. We found that *Angptl2* expression in wild-type mice colon increased following irradiation (Fig EV2G). By day 5, we had observed no difference in weight loss between wild-type and *Angptl2*$^{-/-}$ mice (Fig 3A) after irradiation. Also, we observed no difference in ablation of crypts between genotypes at day 3 (Fig EV2H–K). However, by day 5, *Angptl2*$^{-/-}$ mice displayed marked shortening of the small intestine and colon compared to wild-type mice (Figs 3B and EV2L). Rapid regrowth of intestinal crypts was also prominent in wild-type mice at day 5, whereas some *Angptl2*$^{-/-}$ mice exhibited crypt-less regions at the same time point (Fig 3C–H), suggesting that ANGPTL2 is required for an adequate intestinal regeneration following damage.

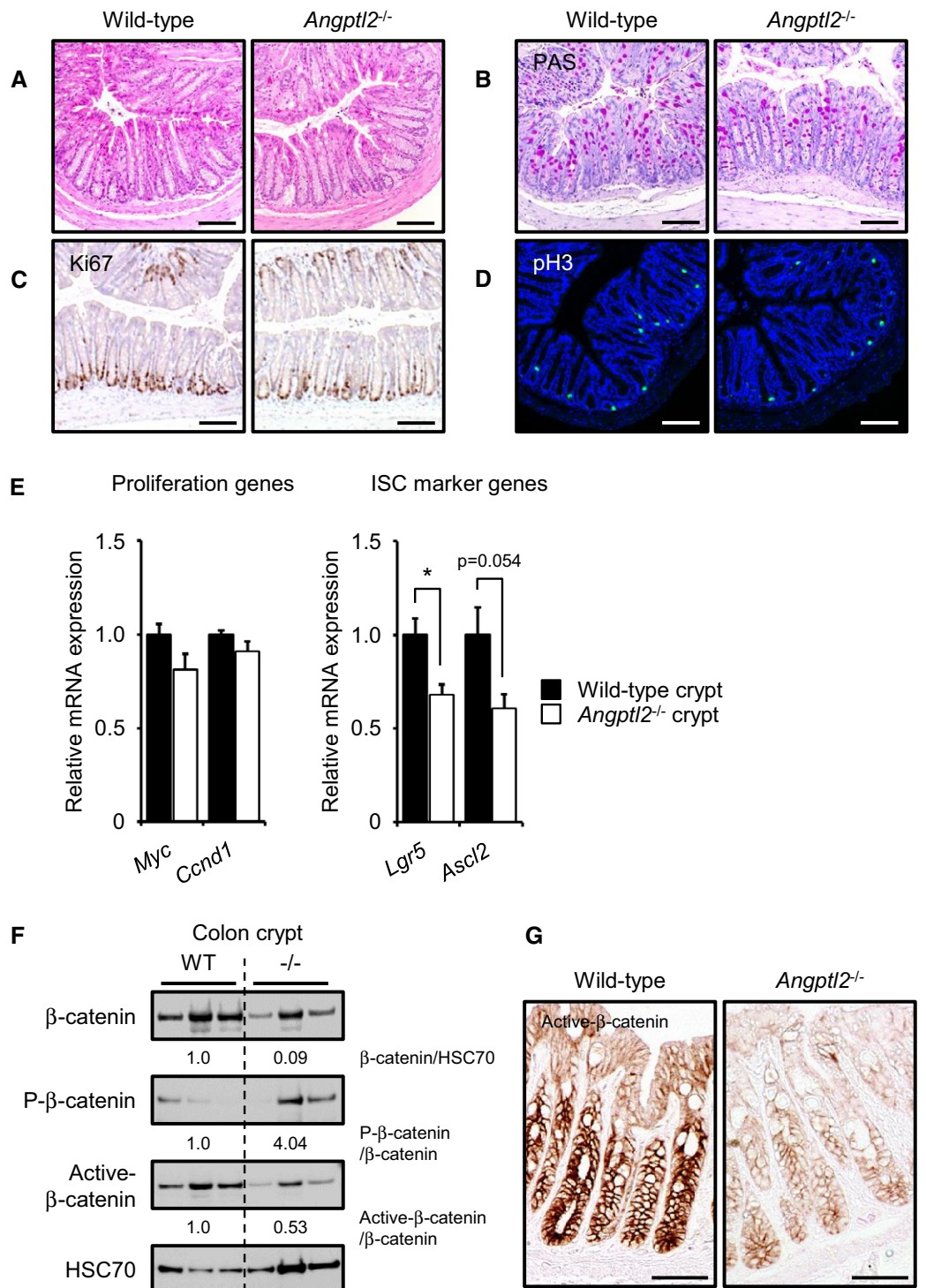

**Figure 1.   The *Angptl2*<sup>−/−</sup> mouse colon shows decreased β-catenin signaling.**

A, B    Representative images of H&E (A)- and PAS (B)-stained colon sections from wild-type and *Angptl2*<sup>−/−</sup> mice. Scale bar = 100 μm.

C, D    Representative images of colon crypts as assessed by Ki67 (C) and pH3 (D) staining. Scale bar = 100 μm.

E    mRNA levels of indicated proliferation and ISC markers in isolated crypts from wild-type (*n* = 4) and *Angptl2*<sup>−/−</sup> (*n* = 4) mice based on qRT–PCR analysis. Data are represented as means ± SEM. *$P < 0.05$, unpaired Student's *t*-test.

F    Western blotting analysis of isolated colon crypts from wild-type and *Angptl2*<sup>−/−</sup> mice. HSC70 served as an internal control. Numbers below panels represent normalized expression of proteins.

G    IHC for expression of active β-catenin in colon or wild-type and *Angptl2*<sup>−/−</sup> mice. Scale bar = 50 μm.

Source data are available online for this figure.

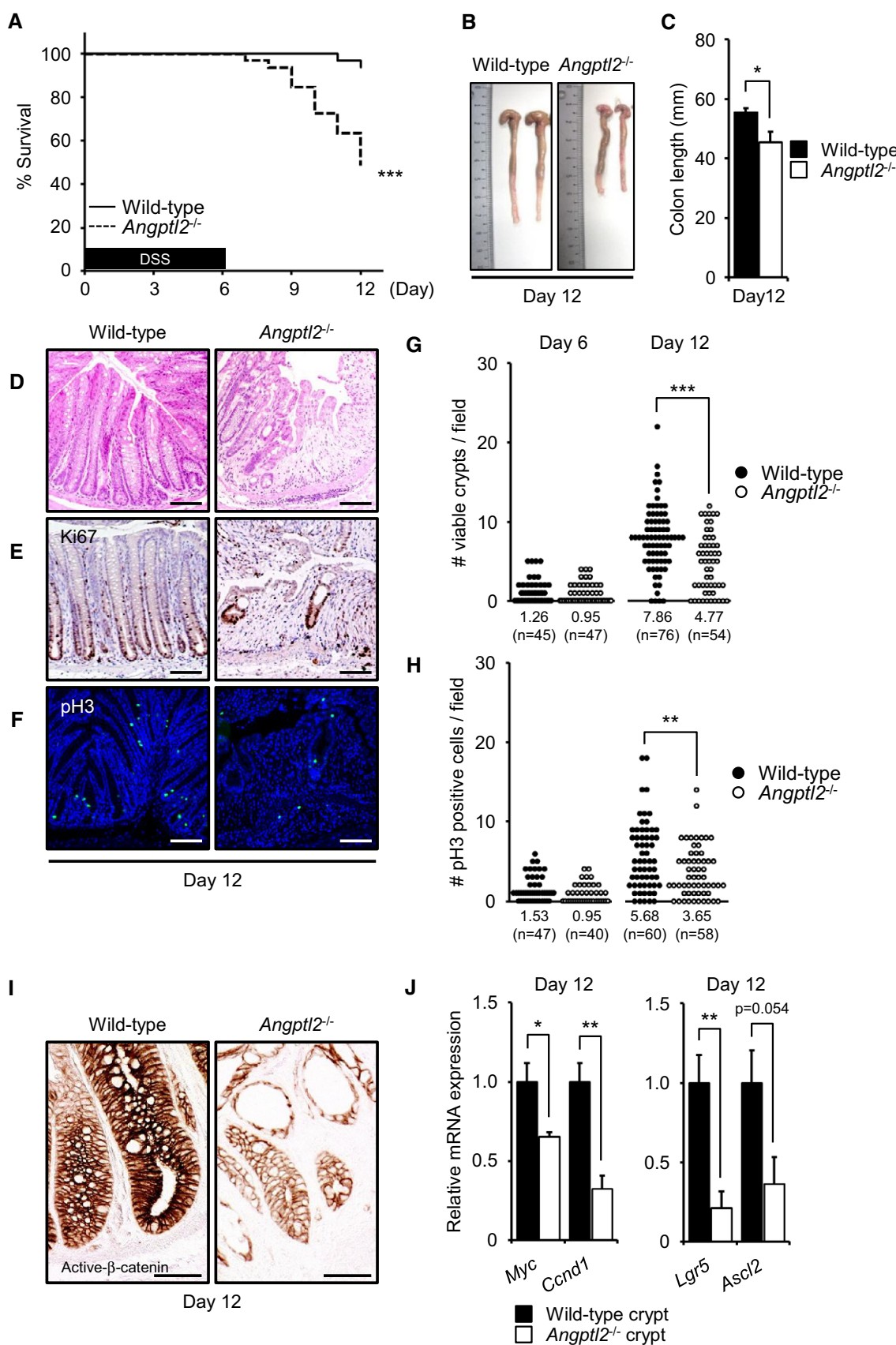

Figure 2.

**Figure 2.   ANGPTL2 functions in intestinal regeneration following DSS treatment.**

A    Survival rates of wild-type (*n* = 33) and *Angptl2*$^{-/-}$ (*n* = 33) mice supplied with drinking water containing 2.5% DSS for 6 days and then untreated water for 6 more days, from two different experiments. ****P* < 0.001, by log-rank test.

B    Representative images of colons from wild-type and *Angptl2*$^{-/-}$ mice following DSS treatment for 6 days and then untreated water for 6 more days (at day 12).

C    Colon length of wild-type (*n* = 7) and *Angptl2*$^{-/-}$ (*n* = 6) mice following DSS treatment for 6 days and then untreated water for 6 more days (at day 12).

D–F   Representative images of colon crypts as assessed by H&E (D), Ki67 (E), and pH3 (F) staining in wild-type and *Angptl2*$^{-/-}$ mice following DSS treatment for 6 days and then untreated water for 6 more days (at day 12). Scale bar = 100 μm.

G, H   The number of regenerating crypts as quantified using Ki67 (G) and pH3 (H) staining. Each data point represents the number of Ki67-positive viable crypts or the number of pH3-positive cells in a single field of view. Multiple areas in at least six mice per genotype were quantified. Numbers below panels represent the average.

I    IHC for expression of active β-catenin in colon or wild-type and *Angptl2*$^{-/-}$ mice following DSS treatment for 6 days and then untreated water for 6 more days (at day 12). Scale bar = 50 μm.

J    mRNA levels of indicated genes in isolated crypts from wild-type (*n* = 4) and *Angptl2*$^{-/-}$ (*n* =4) mice following DSS treatment for 6 days and then untreated water for 6 more days (at day 12).

Data information: Data are represented as mean ± SEM. **P* < 0.05; ***P* < 0.01; ****P* < 0.001.  (C) Unpaired Welch's *t*-test, (G, H) unpaired Student's *t*-test or Welch's *t*-test, (J) unpaired Student's *t*-test.

## ANGPTL2 is localized to intestinal mesenchyme

We next asked where ANGPTL2 is expressed in intestine. *Angptl2* mRNA was widely expressed in various regions of intestine of normal adult mice (Fig 4A). To localize ANGPTL2 protein expression in colon, we performed IHC for ANGPTL2 protein in mice (Fig 4B) and in human normal colon tissue associated with colorectal cancer (Fig EV3A). ANGPTL2 protein was detected in mesenchymal cells, especially in smooth muscle cells (SMCs) and ISEMFs, but ANGPTL2 staining was absent from colonic crypts. qRT–PCR analysis revealed *Angptl2* mRNA transcripts in primary ISEMFs isolated from mouse colon, but *Angptl2* transcripts were undetectable in mouse purified epithelial preparations (Fig 4C). Fluorescent double labeling showed that ANGPTL2 co-localized with alpha-smooth muscle actin (α-SMA)-positive myofibroblasts, but not E-cadherin-positive epithelial cells or CD31-positive endothelial cells (Figs 4D and EV3B).

We previously reported that ANGPTL2 is expressed in macrophages (Horio *et al*, 2014). Thus, we asked whether ANGPTL2 was expressed in bone marrow-derived cells, including macrophages, in inflammatory regions of the colon induced by DSS treatment. IF revealed that ANGPTL2 did not co-localize with markers of macrophages and bone marrow-derived cells (such as CD68-, Ly6G-, and CD45-positive cells) in wild-type mouse colon exposed to DSS in drinking water for 6 days (Fig EV3C). We next employed bone marrow transplantation (BMT) from either wild-type or *Angptl2*$^{-/-}$ mice into recipients of either genotype to determine the contribution of bone marrow-derived cells to DSS-induced colitis. To do so, we undertook bone marrow reconstitution and then 21 days later induced tissue injury by DSS administration for 6 days (Fig EV3D). Mortality after 6 days of DSS treatment between recipients of the same genotype did not differ significantly following BMT with cells of either genotype (Fig EV3E), indicating that ANGPTL2 expression in intestinal mesenchyme rather than bone marrow-derived cells underlies protection against DSS-induced injury.

### *Bmp2* and *Bmp7* are upregulated in *Angptl2*$^{-/-}$ ISEMFs

Intestinal subepithelial myofibroblasts regulate the ISC niche through multiple pathways, including Wnt, BMP, epidermal growth factor (EGF), and insulin-like growth factor (IGF) (Yen & Wright, 2006; Chivukula *et al*, 2014; Ong *et al*, 2014). Thus, we assessed expression of factors functioning in these pathways in cultured

wild-type and *Angptl2*$^{-/-}$ primary ISEMFs (Fig 5A–C). qRT–PCR analysis revealed comparable expression of *Wnt3a*, *Wnt2b*, *Rspo1*, *Egf*, *Fgf*, *Hgf*, *Igf1*, and *Igf2* mRNAs in wild-type and *Angptl2*$^{-/-}$ ISEMFs (Fig 5A and C), but *Bmp2* and *Bmp7* were more highly expressed in *Angptl2*$^{-/-}$ compared to wild-type ISEMFs (Fig 5B). However, we observed no difference in expression of *Bmp4* or of the BMP antagonists *Noggin* or *Grem1* in *Angptl2*$^{-/-}$ compared to wild-type ISEMFs (Fig 5B). Following stem cell differentiation, BMPs inhibit intestinal epithelial proliferation through phosphorylated Smad1/5 proteins (Auclair *et al*, 2007; Hardwick *et al*, 2008). BMP signaling is active at the top of crypt and inhibited at the bottom due to activity of endogenous BMP antagonists (Kosinski *et al*, 2007). Therefore, we employed IHC to examine phosphorylated Smad1/5 (P-Smad1/5) at the crypt bottom in wild-type and *Angptl2*$^{-/-}$ mice treated with or without DSS in drinking water for 6 days on and 6 days off. We found a greater number of P-Smad1/5-positive cells in the lower half of crypts in *Angptl2*$^{-/-}$ mice with or without DSS treatment (Fig 5D and E). These results suggest that BMP/Smads signaling is activated in lower half of crypts in *Angptl2*$^{-/-}$ mice with or without DSS treatment.

Two signaling pathways, Smad1/5 and phosphatase and tensin homolog (PTEN)/β-catenin, function downstream of BMP (Scoville *et al*, 2008). BMP signaling reportedly inhibits ISC renewal by suppressing Wnt/ β-catenin signaling (He *et al*, 2004). BMP signaling also enhances PTEN activity, in turn inhibiting the AKT pathway and destabilizing β-catenin (Tian *et al*, 2005). As β-catenin signaling decreases in the untreated *Angptl2*$^{-/-}$ mouse crypts (Fig 1F and G), we examined potential activation of PTEN-AKT signaling in ISCs in untreated mice. Western blot analysis showed that levels of both the inactive (phosphorylated) form of PTEN (P-PTEN) and the active (phosphorylated) form of AKT (P-AKT) decreased in *Angptl2*$^{-/-}$ relative to wild-type mouse crypts in untreated mice (Fig 5F), suggesting that β-catenin destabilization seen in *Angptl2*$^{-/-}$ mice IECs could be due to AKT inactivation via BMP/PTEN signaling.

### Integrin α5β1/NF-κB signaling downregulates *Bmp* expression

We next asked whether signaling downstream of ANGPTL2 inhibited expression of *Bmp* genes. We previously reported that ANGPTL2 binds to integrin α5β1 (Tabata *et al*, 2009). Thus, we initially assessed expression of integrin α5β1 in colon cells. Both qRT–PCR and IHC analysis revealed that integrin α5β1 was expressed in ISEMFs (Fig 6A and B). We next examined *Bmp* mRNA induction of

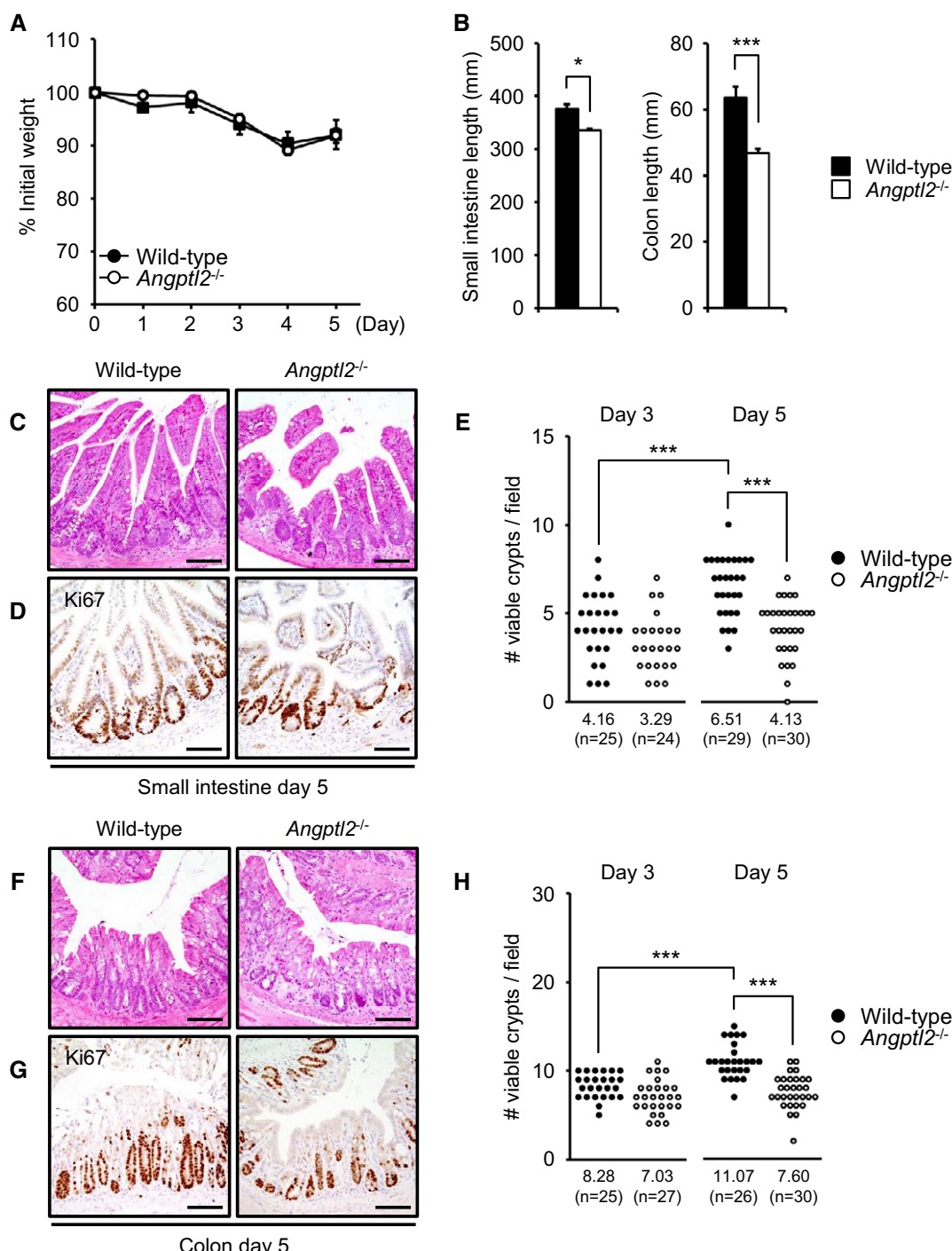

**Figure 3.  ANGPTL2 functions in intestinal regeneration after irradiation.**

A    Weight loss of wild-type (*n* = 4) and *Angptl2*$^{-/-}$ (*n* = 5) mice in days following 12-Gy irradiation.

B    Small intestine and colon length from wild-type (*n* = 4) and *Angptl2*$^{-/-}$ (*n* = 5) mice 5 days after 12-Gy irradiation.

C, D    Representative images of crypts of small intestine as assessed by H&E (C) and Ki67 (D) staining in wild-type and *Angptl2*$^{-/-}$ mice 5 days after 12-Gy irradiation. Scale bar = 100 μm.

E    The number of regenerating crypts in small intestine as quantified by Ki67 (D) staining at days 3 and 5. Each data point represents the number of Ki67-positive viable crypts in a single field of view. Multiple areas in at least three mice per genotype were quantified. Numbers below panels represent average.

F, G    Representative images of colon crypts as assessed by H&E (F) or Ki67 (G) staining in wild-type and *Angptl2*$^{-/-}$ mice 5 days after 12-Gy irradiation. Scale bar = 100 μm.

H    The number of regenerating colon crypts as quantified by using Ki67 (G) staining at days 3 and 5. Each data point represents the number of Ki67-positive viable crypts in a single field of view. Multiple areas in at least three mice per genotype were quantified. Numbers below panels represent average.

Data information: Data are represented as mean ± SEM. *$P$ < 0.05; ***$P$ < 0.001. (B) Unpaired Student's *t*-test, (E, H) unpaired Student's *t*-test.

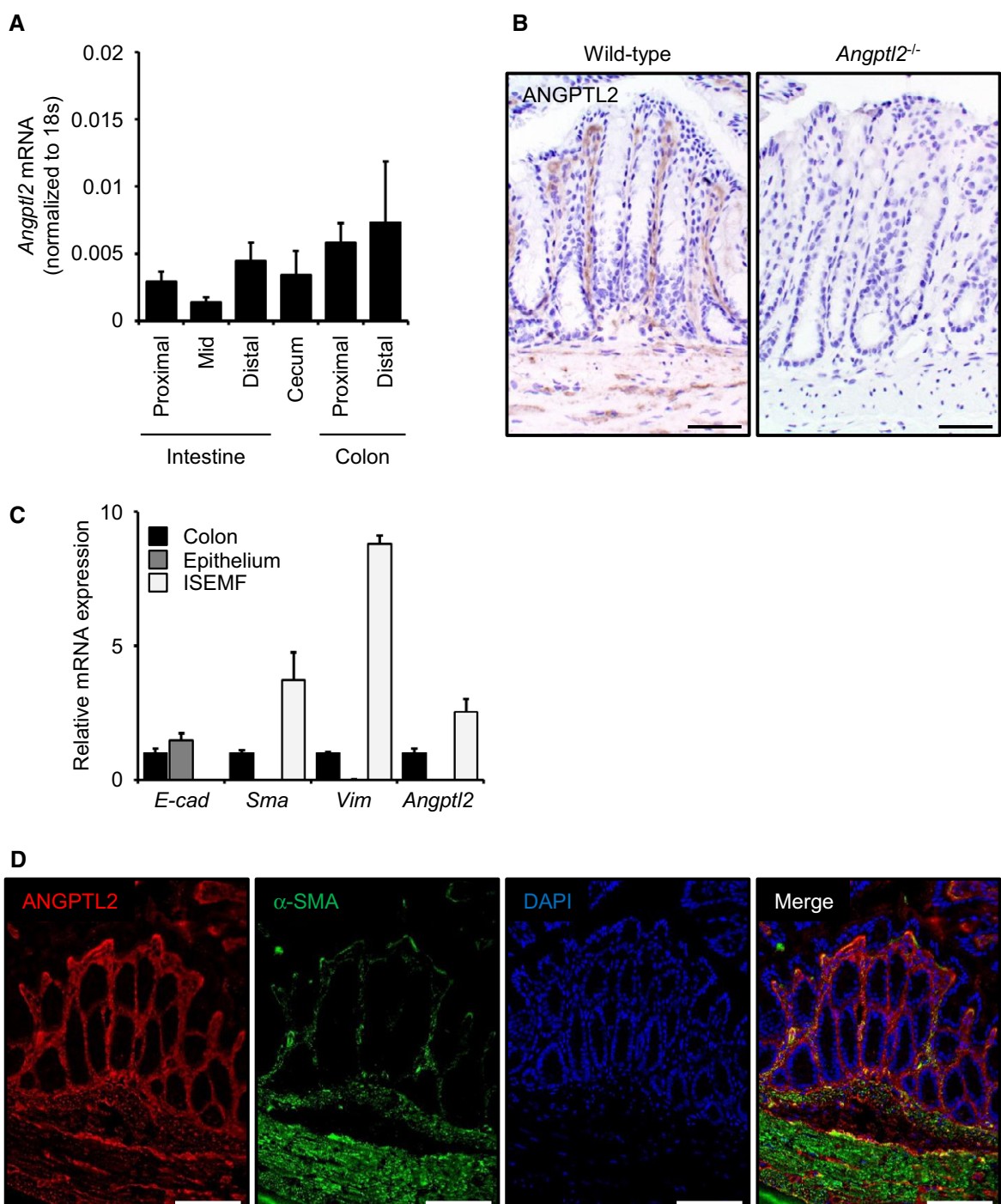

**Figure 4.  ANGPTL2 localizes to the intestinal mesenchyme.**

A   *Angptl2* mRNA levels in small intestine (proximal, mid, and distal), cecum, and colon (proximal and distal) of wild-type mice based on qRT–PCR analysis. n = 5.
B   ANGPTL2 IHC in colon from wild-type and *Angptl2*$^{-/-}$ mice. The *Angptl2*$^{-/-}$ colon image serves as a negative control. Scale bar = 50 μm.
C   mRNA levels of indicated genes in wild-type (n = 4) mice based on qRT–PCR analysis.
D   Representative IF images of distal colon from wild-type mice as assessed with anti-ANGPTL2 (red) and anti-α-SMA (green). Nuclei are counterstained with DAPI (blue). Scale bar = 100 μm.

Data information: Data are represented as mean ± SEM.

cultured wild-type ISEMFs in the presence of a specific function-blocking integrin α5β1 antibody. Both *Bmp2* and *Bmp7* mRNAs were upregulated after treatment of cells with neutralizing antibody for 24 h (Fig 6C). To identify signals potentially downregulating BMPs in this context, we isolated ISEMFs from wild-type mouse colon and treated them with various antagonists of integrin-mediated signaling,

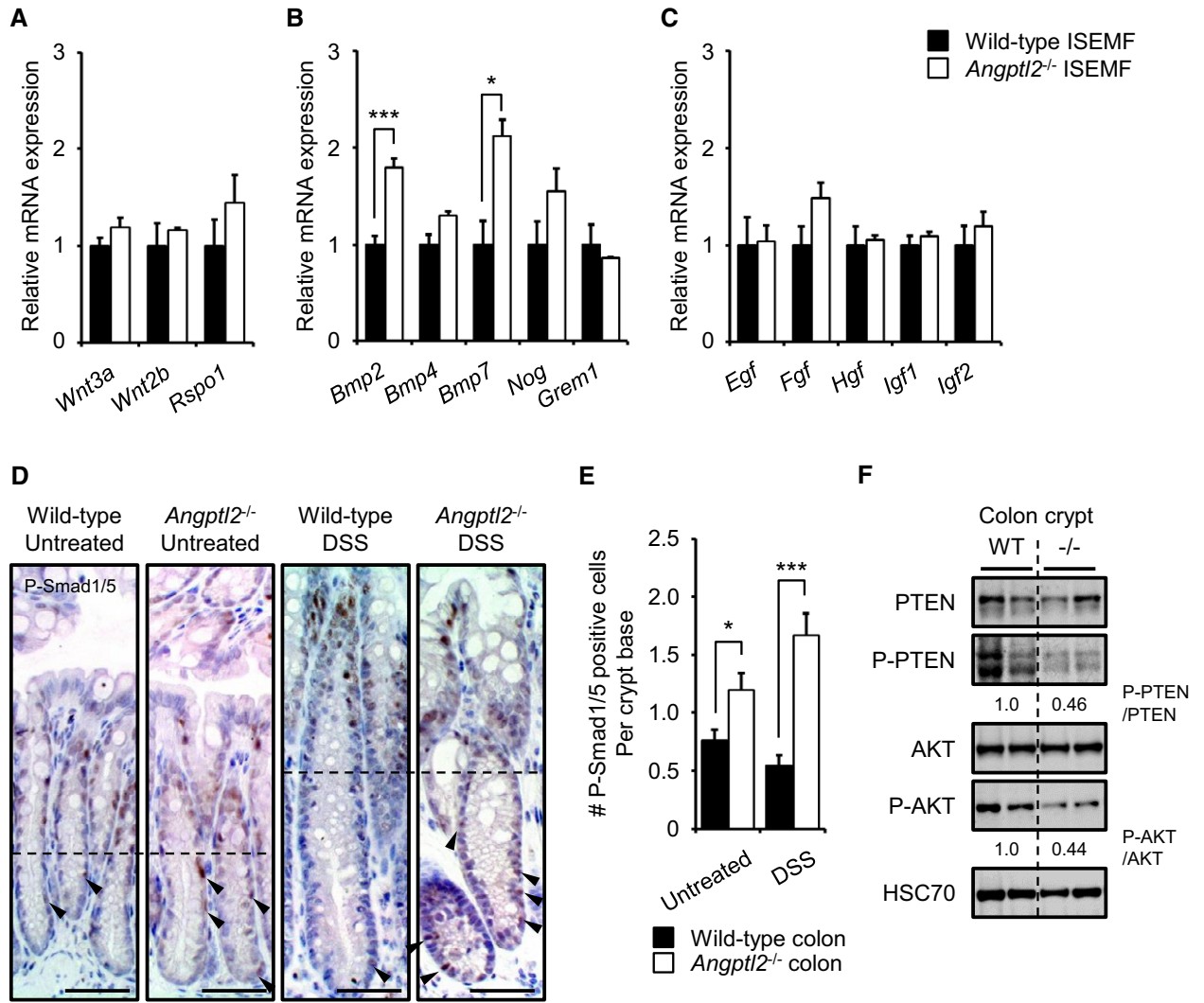

**Figure 5.** *Bmp2* and *Bmp7* transcripts are upregulated in *Angptl2*^−/−^ ISEMFs.

A–C  mRNA levels of indicated genes in ISEMFs from wild-type (*n* = 4) and *Angptl2*^−/−^ (*n* = 3) mice based on qRT–PCR analysis.

D    Representative staining for P-Smad1/5 in crypts from wild-type and *Angptl2*^−/−^ mice following DSS treatment for 6 days and then untreated water for 6 more days (DSS) or untreated water for 12 days (Untreated). Dashed lines indicate half of the crypt, and arrowheads show P-Smad1/5-positive cells in lower halves of crypts. Scale bar = 50 μm.

E    The number of P-Smad1/5-positive cells in the crypt base (*n* = 60) from wild-type and *Angptl2*^−/−^ mice following DSS treatment for 6 days and then untreated water for 6 more days (DSS) or untreated water for 12 days (Untreated). Multiple areas in three mice per genotype were quantified.

F    Western blotting analysis of isolated colon crypts from wild-type and *Angptl2*^−/−^ mice. HSC70 serves as an internal control. Numbers below panels represent normalized protein expression.

Data information: Data are represented as mean ± SEM. *$P < 0.05$; ***$P < 0.001$. (A–C) Unpaired Student's *t*-test, (E) unpaired Welch's *t*-test.
Source data are available online for this figure.

such as extracellular signal-regulated kinase (ERK), p38, c-jun N-terminal kinase (JNK), phosphoinositide 3-kinase (PI3K), and nuclear factor-κB (NF-κB) inhibitors. qRT–PCR analysis revealed that *Bmp2* mRNA induction was slightly upregulated following treatment of cells with the ERK inhibitor U0126 for 24 h and slightly downregulated following treatment with a p38 inhibitor for 24 h. *Bmp7* mRNA induction was slightly downregulated following treatment of cells with either a p38 or a JNK inhibitor for 24 h. Induction of *Bmp2* and *Bmp7* mRNA, however, was significantly upregulated following treatment of cells with the NF-κB inhibitor BAY11-7085 for

24 h (Fig 6D), suggesting that integrin α5β1/NF-κB signaling down-regulates *Bmp* expression. These results suggest that ANGPTL2 derived from ISEMFs might inhibit *Bmp2* and *Bmp7* mRNA induction mainly through integrin α5β1/NF-κB signaling pathway.

### ANGPTL2 does not directly activate β-catenin signaling in epithelial cells *in vitro*

Given that ANGPTL2 is secreted, we asked whether ANGPTL2 derived from ISEMFs directly activated β-catenin signaling pathway in

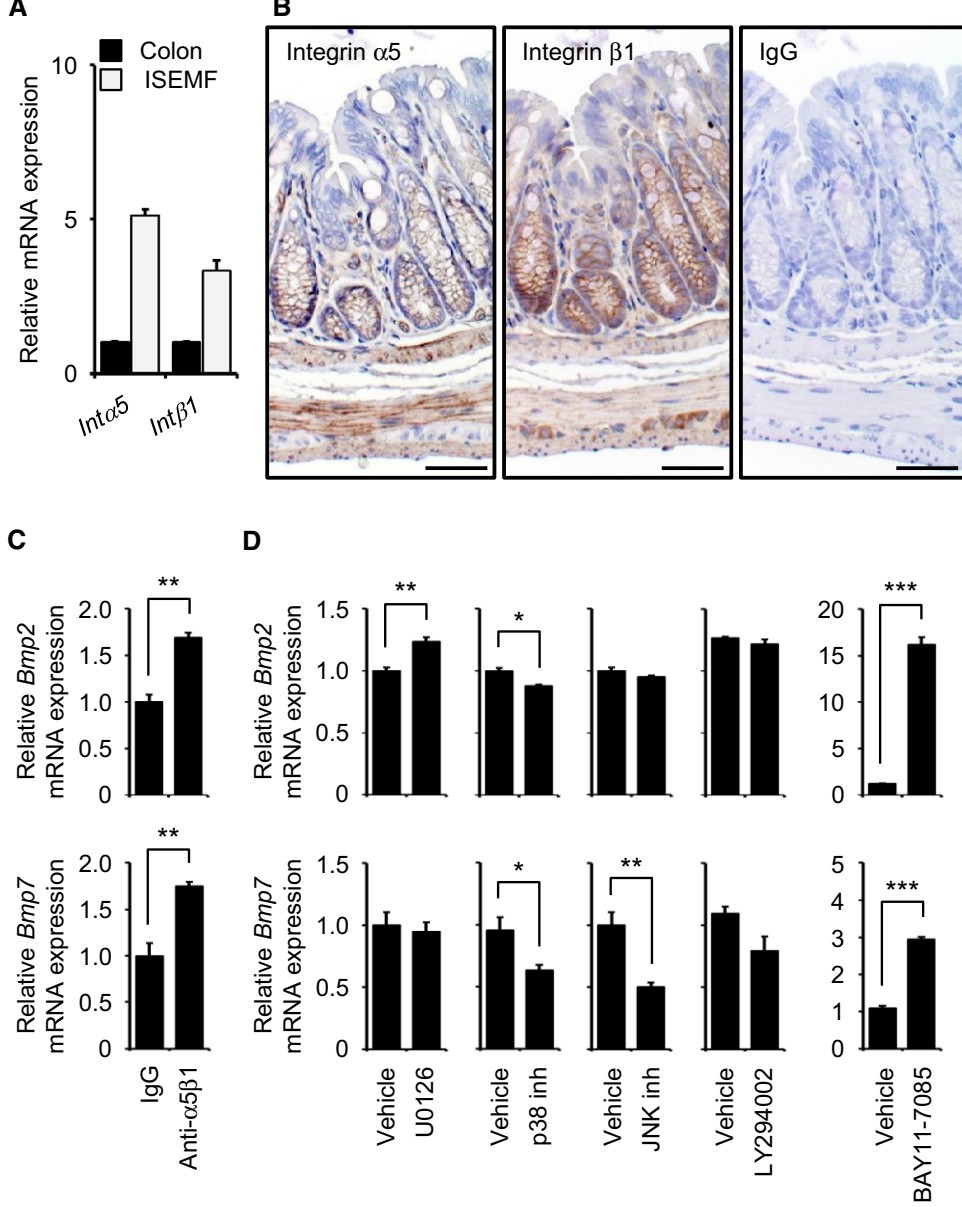

**Figure 6. Integrin α5β1/NF-κB signaling downregulates *Bmp* expression.**

A    mRNA levels of indicated genes in colon and cultured ISEMFs from wild-type mice (n = 4) based on qRT–PCR analysis. *Intα5*, integrin α5; *Intβ1*, integrin β1.
B    Representative IHC of integrin α5β1 in colon from wild-type mouse. Scale bar = 50 μm.
C, D    mRNA levels of *Bmp2* and *Bmp7* in ISEMFs from wild-type mice 24 h after treatment with integrin α5β1 antibody or various signaling pathway inhibitors. n = 3. U0126, ERK inhibitor; LY294002, PI3K inhibitor; BAY11-7085, NF-κB inhibitor.

Data information: Data are represented as mean ± SEM. *P < 0.05; **P < 0.01; ***P < 0.001. (C) Unpaired Student's *t*-test, (D) unpaired Student's *t*-test or Welch's *t*-test.

epithelial cells in a paracrine manner. β-Catenin stabilization is reportedly regulated by integrin signaling (Hannigan *et al*, 2005; Marie *et al*, 2014). Therefore, we hypothesized that ANGPTL2 derived from ISEMFs directly regulates β-catenin in crypts through integrin α5β1 signaling. To determine the hypothesis, we undertook organoid culture using crypts isolated from wild-type mice (Sato *et al*, 2011a) treated with or without recombinant ANGPTL2 (rANGPTL2) protein. We did not observe a growth and size effect on cultures following rANGPTL2 treatment under optimized culture conditions that included

Wnt supplementation (Wnt (+); Fig 7A and B). We also observed no difference in formation of colon crypts with or without rANGPTL2 protein under culture conditions lacking Wnt supplementation (Wnt (−); Fig 7A and B). To determine a potential role for rANGPTL2 protein in organoid-forming capacity, we delivered a damaging blow to organoids by two methods: by breaking them up through passage through a syringe (Fig EV4A) or by irradiation (Fig EV4B). Although the number of viable organoids decreased after either injury, organoid-forming capacity was comparable with or without rANGPTL2

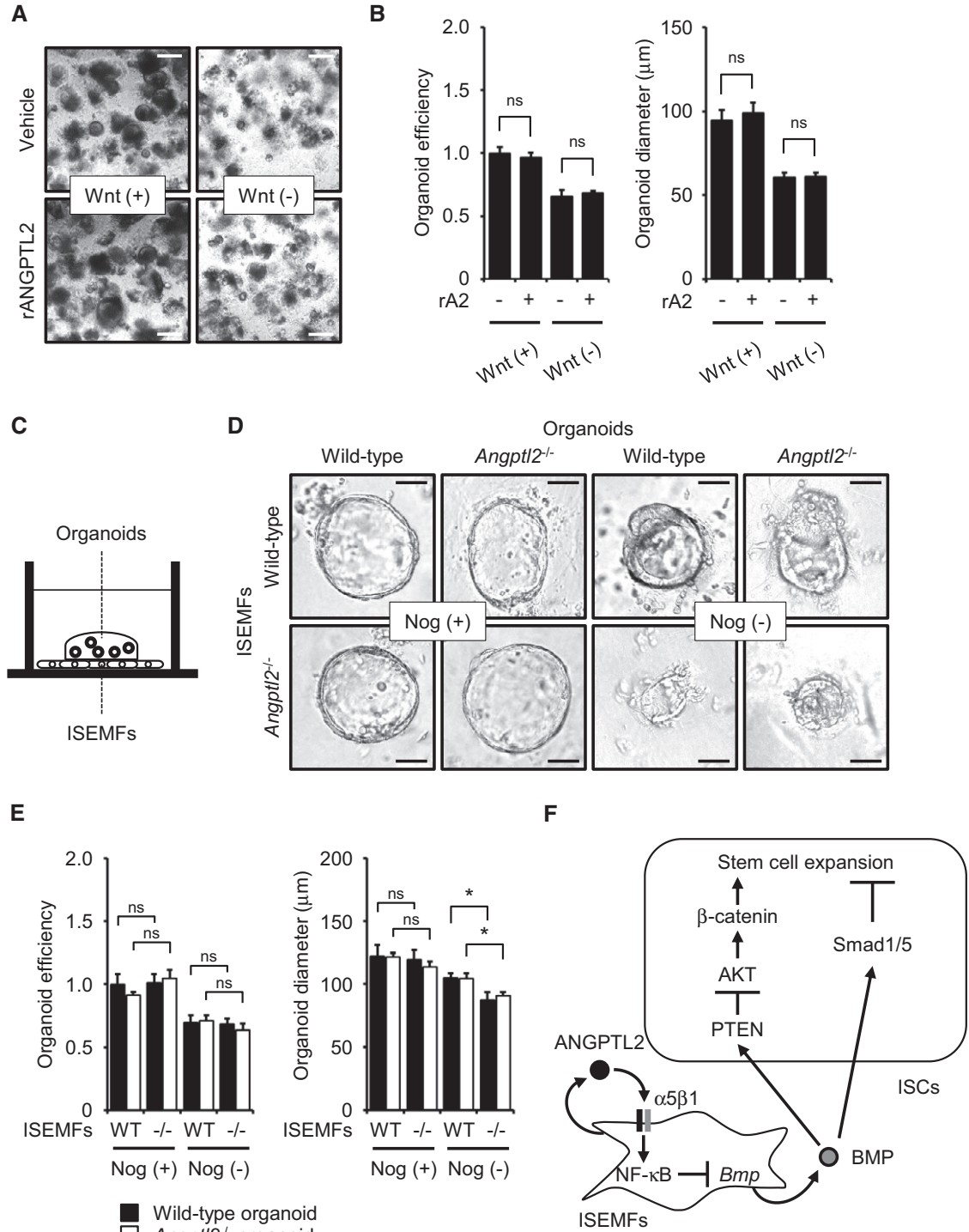

**Figure 7.  Co-culture with *Angptl2*$^{-/-}$ ISEMFs decreases organoid formation *in vitro*.**

A   Colon organoid cultures treated with vehicle or rANGPTL2 protein. Scale bar = 200 μm.

B   Organoid growth efficiency and size following treatment with vehicle or rANGPTL2. rA2, recombinant ANGPTL2. *n* = 4. Data from vehicle/Wnt (+) were set at 1.

C   Schematic illustration of co-culture of colon organoids with ISEMFs in direct contact.

D   Colon organoid cultures in the presence of ISEMFs with or without Noggin (Nog). Scale bar = 50 μm.

E   Organoid growth efficiency and size in the presence of ISEMFs with or without Noggin (Nog). *n* = 4. Data from wild-type organoids/wild-type ISEMFs/Nog (+) were set at 1.

F   Model of ANGPTL2 activity in the intestinal stem cell niche. ISEMF-derived ANGPTL2 inhibits BMP signaling in an autocrine manner through the integrin α5β1/NF-κB pathway to maintain ISC stemness by regulating β-catenin and Smad 1/5 signaling.

Data information: Data are represented as mean ± SEM. *$P$ < 0.05; ns, not statistically significant. (B) Unpaired Student's *t*-test, (E) unpaired Student's *t*-test.

treatment (Fig EV4C–F), suggesting that ANGPTL2 derived from ISEMFs does not directly affect IEC activity. We next examined whether rANGPTL2 protein directly activated β-catenin signaling in epithelial cells using the human colon carcinoma lines Caco-2 and SW480 and the human embryonic kidney cell line (HEK293), as it was technically difficult to obtain a sufficient number of epithelial cells from organoid culture. We initially confirmed that these cell lines expressed integrin α5β1 (Fig EV4G and H). Treatment of any of these three lines with rANGPTL2 protein did not alter expression β-catenin or P-β-catenin or β-catenin, PTEN or AKT activity (Fig EV4I).

### Organoid formation *in vitro* decreases following co-culture with *Angptl2$^{-/-}$* ISEMFs

The formation of organoids from isolated crypts may mimic conditions similar to tissue regeneration (Date & Sato, 2015; Chiacchiera *et al*, 2016; Oudhoff *et al*, 2016). To assess the function of *Angptl2$^{-/-}$* ISEMFs in intestinal regeneration, we assessed organoid formation in a co-culture system. To do so, we co-cultured wild-type or *Angptl2$^{-/-}$* mouse crypts directly on top of a confluent mono-layer of ISEMFs derived from mice of either genotype (Fig 7C). We observed no difference in organoid growth or size in the presence of wild-type or *Angptl2$^{-/-}$* ISEMFs under optimized culture conditions, including those containing exogenous Noggin supplementation (Nog (+); Fig 7D and E). However, we reasoned that activity of BMP derived from ISEMFs might be antagonized by either exogenous or ISEMF-derived endogenous Noggin. Therefore, we undertook experiments without exogenous Noggin supplementation. In those conditions, we observed decreased size of organoids of either genotype cultured in the presence of *Angptl2$^{-/-}$* relative to wild-type ISEMFs (Nog (−); Fig 7D and E).

Next, we co-cultured wild-type or *Angptl2$^{-/-}$* mouse crypts with ISEMFs using Transwell chambers, with crypts cultured in the upper chamber and ISEMFs cultured in the lower (Fig EV5A). In this experimental system, we also observed decreased size of organoids cultured in the presence of *Angptl2$^{-/-}$* relative to wild-type ISEMFs under Nog (−) conditions (Fig EV5B and C). Finally, we treated organoids with conditioned medium (CM) from ISEMFs (Fig EV5D). The size of crypts treated with *Angptl2$^{-/-}$* ISEMF CM slightly decreased relative to those treated with wild-type ISEMF CM (Fig EV5E and F). These results suggest overall that impaired regeneration seen in *Angptl2$^{-/-}$* mouse crypts is due to upregulation of ISEMF-derived BMP expression (Fig 7F).

## Discussion

We have shown that aberrant ANGPTL2 signaling causes chronic inflammatory diseases, such as obesity, metabolic disease, type 2 diabetes, atherosclerotic disease, and possibly some cancers (Oike & Tabata, 2009; Tabata *et al*, 2009; Aoi *et al*, 2011; Endo *et al*, 2012; Horio *et al*, 2014). Therefore, ANGPTL2 does not seem to have a developmental role, but instead functions in pathological conditions. Interestingly, zebrafish *Angptl2* mRNA expression is reportedly induced abundantly during fin regeneration in blastema tissue, suggesting a role for ANGPTL2 in tissue repair (Kubota *et al*, 2005b). Here, we found that *Angptl2* expression in wild-type mouse colon increases following DSS treatment and irradiation, and we

showed that ANGPTL2 functions in intestinal regeneration. In this study, we show that ISEMF-derived ANGPTL2 is important to regulate epithelial wound repair in models of DSS-induced colitis. In untreated conditions, ISEMFs expressed higher levels of BMP in *Angptl2$^{-/-}$* relative to wild-type mice, inactivating β-catenin signaling to decrease *Lgr5* mRNA induction in IECs. ANGPTL2 derived from ISEMFs maintained the ISC niche by modulating levels of competing signaling between BMP and β-catenin to maintenance ISCs in physiologic condition, suggesting that intestinal regeneration after injury was significantly impaired by perturbed β-catenin/BMP signaling in *Angptl2$^{-/-}$* mice. These results suggest that the ISC niche maintained by ISEMF-derived ANGPTL2 regulates intestinal homeostasis after structural or functional tissue damage induced by either external or internal stress (Appendix Fig S1).

Environmental influences on the ISC niche are likely complex. Intestinal stem cells reside near the bases of crypts (Barker *et al*, 2007), and many local constituents likely function in regeneration or maintenance of homeostasis after injury, including ISEMFs, immune cells, endothelial cells, and luminal microbiota. All may regulate ISCs (Walker *et al*, 2009), but ISEMFs may also govern epithelial cell functions, such as proliferation, differentiation, and/or extracellular matrix metabolism, or may alter the growth of basement membrane through epithelial–mesenchymal crosstalk (Andoh *et al*, 2005; Yen & Wright, 2006). ISEMFs differentiate from SMCs, proto-myofibroblasts, and bone marrow-derived cells (Powell *et al*, 2011). Here, we found that ANGPTL2 is expressed in ISEMFs and muscularis mucosa as well as muscular layer. Our BMT experiments indicate that ANGPTL2 derived from bone marrow cells is not required for intestinal regeneration in wild-type or *Angptl2$^{-/-}$* mice. On the other hand, ISEMFs differentiated from SMCs are reportedly critical regulators of intestinal regeneration (Chivukula *et al*, 2014). Therefore, we suggest that ISEMFs differentiated from SMCs in muscularis mucosa regulate intestinal regeneration and homeostasis via ANGPTL2 expression.

Intestinal homeostasis is regulated by polarized signaling of BMPs and Wnt/β-catenin. Injury-induced BMP signaling reportedly negatively regulates intestinal stem cell activity, regeneration, and homeostasis through Smad signaling (Guo *et al*, 2013; Ayyaz *et al*, 2015). BMP signaling limits stem cell self-renewal and specifies fate of proliferating progenitor cells (Wakefield & Hill, 2013). At the crypt base, stem cell expansion and TA cell proliferation are driven by high β-catenin and low BMP levels, whereas daughter cell differentiation and apoptosis are controlled by low β-catenin and high BMP at the luminal surface (Wakefield & Hill, 2013). To maintain these gradients, BMP antagonists such as Noggin, GREM1, or GREM2 are expressed in ISEMFs within the crypt base stem cell niche, where they override BMP signaling to permit Wnt/β-catenin-driven stem cell self-renewal (Scoville *et al*, 2008; Reynolds *et al*, 2014; Worthley *et al*, 2015). Here, we show that *Angptl2$^{-/-}$* ISEMFs upregulate *Bmp2* and *Bmp7* mRNAs, but not *Noggin* or *Grem1* transcript levels. In organoid co-culture systems, we observed no difference in growth or size of organoids in the presence of either wild-type or *Angptl2$^{-/-}$* ISEMFs under culture conditions containing exogenous Noggin. However, in the absence of exogenous Noggin, we observed decreased size of organoids co-cultured with *Angptl2$^{-/-}$* ISEMFs relative to those co-cultured with wild-type ISEMFs. These results suggest that in the absence of exogenous Noggin, activity of BMP secreted from *Angptl2$^{-/-}$* ISEMFs may

predominate over that of Noggin derived from ISEMFs. On the other hand, organoids size in the presence of *Angptl2*$^{-/-}$ ISEMF CM decreased relative to that seen in the presence of wild-type ISEMF CM, with or without exogenous Noggin. In these experiments, BMP secreted from *Angptl2*$^{-/-}$ ISEMFs may overcome the antagonizing capacity of Noggin, as Noggin levels would be reduced.

We also observed increased *Bmp2* and *Bmp7* mRNA induction in cultured ISEMFs following treatment with function-blocking integrin α5β1 antibodies or the NF-κB inhibitor BAY11-7085. ANGPTL2 reportedly activates NF-κB through integrin α5β1 in fibroblasts (Nakamura *et al*, 2015), suggesting that ANGPTL2 derived from ISEMFs suppresses *Bmp2* and *Bmp7* mRNA induction through integrin α5β1/NF-κB signaling. On the other hand, ERK inhibitor U0126 treatment of rat mesenchymal stem cells reportedly increases *Bmp2* induction to activate BMP/Smad signaling (Xu *et al*, 2015). We also previously reported that ANGPTL2 activates ERK through integrin α5β1 in chronic kidney disease (Morinaga *et al*, 2016). Here, we showed that induction of *Bmp2* transcripts is upregulated by the ERK inhibitor U0126. Thus, NF-κB and/or ERK signaling promoted by ANGPTL2 might repress *Bmp2* mRNA induction in regulating intestinal homeostasis. On the other hand, the transcription factor GATA6 reportedly represses *Bmp* expression (Whissell *et al*, 2014). Thus, regulation of *Bmp* expression merits further investigated.

Organoid formation from isolated crypts reportedly mimics conditions of tissue regeneration (Date & Sato, 2015; Chiacchiera *et al*, 2016; Oudhoff *et al*, 2016). Accordingly, we feel that our comparable analysis of colonic organoid formation in the presence or absence of recombinant ANGPTL2 protein serves as a useful system in which to test a potential paracrine role for ANGPTL2 in regulating intestinal regeneration. We observed no difference in formation of colonic organoids from isolated crypts with or without recombinant ANGPTL2 protein, suggesting that ISEMF-derived ANGPTL2 does not directly affect IEC regenerative responses.

We and others have reported that ANGPTL2 activates PI3K/AKT and PI3K/NF-κB signaling (Kubota *et al*, 2005a; Horiguchi *et al*, 2014), and we proposed that ANGPTL2 might activate AKT/β-catenin through PI3K, even if it does not alter PTEN activity. However, following *in vitro* analysis of various cell lines reported here, we found that β-catenin, PTEN, and AKT activities remained unchanged by rANGPTL2 treatment. In organoid culture, rANGPTL2 protein did not alter survival of colon crypts in the presence or absence of Wnt, suggesting that unknown factors are required for ANGPTL2 protein to affect these cells. These possibilities should be tested in future studies.

Wnt/β-catenin signaling controls epithelial proliferation, intestinal homeostasis, and ISC maintenance (Sancho *et al*, 2003; Kuhnert *et al*, 2004). On the other hand, abnormal activation of β-catenin leads to adenomas and colorectal cancer development (Miyoshi *et al*, 1992; Kinzler & Vogelstein, 1996). BMP/Smad pathway is known to be tumor suppressor in colorectal cancer (Beck *et al*, 2006; Kodach *et al*, 2008; Allaire *et al*, 2016). We report de-repression of *Bmp2* and *Bmp7* in *Angptl2*$^{-/-}$ ISEMFs and decreased accumulation of β-catenin in *Angptl2*$^{-/-}$ mouse crypts, suggesting that ANGPTL2 may function as a tumor promotor in intestine. Abundant ANGPTL2 expression is reportedly highly correlated with frequency of carcinogenesis and tumor metastasis (Aoi *et al*, 2011, 2014) in lung cancer (Endo *et al*, 2012), breast cancer (Masuda *et al*, 2015), colorectal cancer (Toiyama *et al*, 2014), and osteosarcoma (Odagiri *et al*,

2014). In addition, *ANGPTL2* is reportedly upregulated in tumor-associated fibroblasts (TAFs) from anti-VEGF-resistant tumors (Crawford *et al*, 2009), suggesting that fibroblast-derived ANGPTL2 functions in tumor development. The relationship between ANGPTL2 expression and intestinal tumors warrants further investigation.

# Materials and Methods

### Mice

*Angptl2*$^{-/-}$ mice were described previously (Tabata *et al*, 2009). All experimental protocols were approved by the Ethics Review Committee for Animal Experimentation of Kumamoto University. Age- and sex-matched mice were used for the experiments.

### DSS treatment and irradiation

For treatment, 2.5% (w/v) DSS of molecular weight 36,000–50,000 (MP Biochemicals) was supplied in drinking water for 6 days, and then, DSS was replaced by normal water for 6 more days. Body weight, stool consistency, and occult/gross fecal bleeding were assessed and scored to obtain a disease activity index (DAI) (Table EV1). All animals were evaluated daily. The final index was the mean of these three noted scores. Occult blood in feces was evaluated using a Uropaper III test (Eiken chemical Co., Ltd.). For the regeneration model after irradiation, mice were irradiated (12 Gy) using the Gammacell 40 Exactor (Best Theratronics), and body weight was evaluated daily. Mice were euthanized and intestinal repair was assessed at days 3 and 5.

### Histology, immunohistochemistry, and immunofluorescence

Intestinal segments were dissected, washed in ice-cold PBS, and fixed for 24 h at room temperature (RT) in 15% neutral buffered formalin, and 4-μm paraffin sections were prepared using a standard protocol. For mouse ANGPTL2 IHC, frozen OCT-embedded tissue sections were used. Routine H&E staining and PAS staining were performed for general histology. For IHC, after antigen retrieval, endogenous peroxidase was blocked using 3% H$_2$O$_2$ for 10 min. Samples were blocked with 5% serum for 20 min at RT and then were incubated with primary antibodies overnight at 4°C. Appropriate secondary antibodies were applied for 60 min at RT; 0.02% DAB solution was used for detection and visualization of staining. Slides were counterstained with hematoxylin and mounted. For IF, samples were blocked with 5% serum for 20 min at RT and then were incubated with primary antibodies overnight at 4°C. Secondary antibodies conjugated Alexa Fluor 488 or Alexa Fluor 594 were diluted 1:300 and used for 60 min at RT. Nuclei were stained with DAPI for 20 min. Antibodies used were as follows: anti-Ki67 (1:800, Leica Microsystems, #NCL-Ki67p), anti-phosphorylated histone H3 (1:200, Cell Signaling Technology, #9701), anti-α-SMA (1:4, DAKO, #U7033), anti-active β-catenin (1:200, Cell Signaling Technology, #19807), anti-E-cadherin (1:100, BD, #610182), anti-CD31 (1:100, BD, #557355), anti-CD68 (1:100, AbD Serotec, #MAC1957), anti-Ly6G (1:100, Abcam, #ab25377), anti-CD45 (1:150, BD, #550539), anti-P-Smad1/5 (1:50, Cell Signaling Technology, #9516), anti-integrin α5 (1:100, Abcam, #ab150361), and anti-integrin β1

(1:1,000, Abcam, #ab179471). IHC for mouse ANGPTL2 was performed as described (Motokawa *et al*, 2016) and for human ANGPTL2 as described in Endo *et al* (2012). BrdU pulse-chase experiments were performed with BrdU (SIGMA). Mice were injected i.p. with 50 mg/kg of BrdU before sacrifice. BrdU IHC was performed with staining kit (BD) according to the manufacturer's protocol.

**Colonic crypt isolation, culture, and treatment**

Mouse crypts were isolated as described (Andersson-Rolf *et al*, 2014) with minor modifications. In brief, the colon was cut longitudinally and minced into 5-mm pieces. Tissue was then washed twice with cold PBS and incubated in 5 mM EDTA in PBS for 90 min at 4°C on a tube roller. After incubation, small pieces were vigorously shaken and passed through a 100-μm cell strainer and centrifuged at 300 *g* for 5 min. The collected crypts were used for culture, immunoblotting, or qRT–PCR. For culture, isolated crypts were embedded in Matrigel (Falcon), followed by seeding on 24-well plate. After Matrigel polymerization, crypt culture medium (Advanced DMEM/F12 supplemented with penicillin/streptomycin, GlutaMAX, N2, B27, and *N*-acetylcysteine) containing indicated growth factors (Wnt conditioned medium, 50 ng/ml EGF (Pepro-Tech), 100 ng/ml Noggin (PeproTech), and 1 μg/ml R-spondin1 (PeproTech)) was overlaid; 10 μg/ml rANGPTL2 protein (Yugami *et al*, 2016) or 0.0001 vol.% acetic acid (solvent control) was contained in this culture medium at starting of organoid culture. Medium with or without rANGPTL2 was changed every 2 days. For the regeneration model, 3 days after plating, organoids were irradiated (1 Gy) using the Gammacell 40 Exactor and then assessed 2 days later. Organoid efficiency was determined by counting the number of viable organoids.

**ISEMF isolation and culture**

Mouse ISEMFs were isolated as described (Khalil *et al*, 2013) with minor modifications. The colon was washed with ice-cold PBS and shaken for 90 min in PBS containing 5 mM EDTA at 4°C on a tube roller. Tissue was then incubated in 20 ml RPMI-5 (RPMI with 5% FCS, 10 mM HEPES, 2 mM L-glutamine, 1 mM sodium pyruvate, penicillin/streptomycin, and 2-mercaptoethanol) containing 10.5 mg of dispase and 7.2 mg of collagenase D for 2 h in a shaking 37°C incubator. Digested tissue was treated with ACK lysis buffer (0.15 M $NH_4Cl$, 10 mM $KHCO_3$, and 1 mM EDTA, pH 7.2–7.4) for 5 min and then passed through a 100-μm cell strainer into 100-mm dishes of RPMI-5. After 3-h incubation, non-adherent cells were washed away. One week later, only myofibroblasts divided, and macrophages and epithelial cells had senesced after the first passage.

**Organoid co-culture with ISEMF**

Organoids and ISEMFs were co-cultured as described (Lei *et al*, 2014) with minor modifications. In brief, 250 crypts in 20 μl Matrigel were plated on top of a confluent ISEMF monolayer. For Transwell co-cultures, 250 crypts in 20 μl Matrigel were cultured on the Transwell cell culture membrane (Corning) with a confluent ISEMF layer in the well below. ISEMF CM was collected from confluent ISEMFs in monoculture after 3 days of incubation. Crypts grown with ISEMF CM were given media composed of 50% ISEMF CM and 50% crypt culture medium, plus Noggin and R-spondin1 at indicated concentrations.

**Total RNA extraction and real-time quantitative RT–PCR**

Total RNA was isolated from tissues and cells using TRIzol reagent (Invitrogen). DNase-treated RNA was reverse-transcribed with a PrimeScript RT reagent Kit (Takara Bio). PCR products were analyzed using a Thermal Cycler Dice Real Time System (Takara Bio), and relative transcript abundance was normalized to that of 18S mRNA. Oligonucleotides used for PCR are listed in Table EV2.

**Immunoblot analysis and antibodies**

Tissues and cells were homogenized. Solubilized proteins in lysis buffer (0.1 M Tris–HCl (pH 6.8), 4% SDS, 20% glycerol, 12% 2-mercaptoethanol, and BPB) were subjected to SDS–PAGE, and proteins were electrotransfered to nitrocellulose membranes. Immunodetection was carried out using an ECL kit (GE Healthcare) according to the manufacturer's protocol. Antibodies used were as follows: anti-β-catenin (1:5,000, BD, #610154), anti-P-β-catenin (1:2,000, Cell Signaling Technology, #4176), anti-active-β-catenin (1:2,000, Cell Signaling Technology, #19807), anti-PTEN (1:2,000, Cell Signaling Technology, #9559), anti-P-PTEN (1:2,000, Cell Signaling Technology, #9554), anti-AKT (1:1,000, Cell Signaling Technology, #9272), anti-P-AKT (1:1,000, Cell Signaling Technology, #9271), anti-IGF-1Rβ (1:1,000, Cell Signaling Technology, #3018), anti-P-IGF-1Rβ (1:1,000, Cell Signaling Technology, #4568), and anti-HSC70 (1:2,000, Santa Cruz Biotechnology, #sc7298).

**Cell culture**

HEK293, Caco-2, and SW480 cell lines, purchased from the American Type Culture Collection (ATCC), were cultured in DMEM (Wako) for HEK293 and Caco-2 or Leibovitz's medium (Invitrogen) for SW480, in all cases supplemented with 10% fetal calf serum (FCS) at 37°C in a humidified 5% $CO_2$ atmosphere.

**Bone marrow transplantation (BMT)**

Mouse bone marrow transplantation procedures have been described (Horio *et al*, 2014). In brief, recipient mice at 6 weeks of age underwent 9-Gy total body irradiation using the Gammacell 40 Exactor to eradicate BM cells and then received BM cells from donor mice intravenously. Twenty-one days later, recipients were treated with 2.5% DSS in drinking water for 6 days followed by regular water for 6 days thereafter.

**Flow cytometry analysis**

Cells were incubated with 10 μg/ml anti-α5β1 (JBS5), anti-αvβ3 (LM609), anti-αvβ5 (P1F6) (all from Millipore, Temecula, CA, USA), or respective isotype-matched control IgG for 30 min at 4°C. After washing, cells were incubated with 10 μg/ml Alexa 488-conjugated IgG for another 30 min at 4°C, washed twice, and analyzed by FACS using BD Accuri C6 software (BD) and FlowJo software.

## Human samples

Paraffin-embedded tumor samples were obtained from biopsy specimens of primary tumor from patients ($n = 3$) with unresectable colorectal cancer. This study was approved by the Ethics Committee of Kumamoto University. Written informed consent was obtained from each subject.

## Statistical analysis

Data are represented as means ± SEM. The survival rate was analyzed by log-rank test using GraphPad Prism5 software. A two-tailed Student's *t*-test was performed to calculate *P*-values, unless specified otherwise. Results with *P*-values < 0.05 were considered significant.

**Expanded View** for this article is available online.

## Acknowledgements

We thank Mss. K. Tabu, M. Nakata, and N. Shirai for technical assistance. This work was supported by the Scientific Research Fund of the Ministry of Education, Culture, Sports, Science and Technology (MEXT) of Japan (Grants 15H01520 and 15H06508), by the Core Research for Evolutional Science and Technology (CREST) program of the Japan Science and Technology Agency (JST) (Grant 13417915), and by the CREST program of the Japan Agency for Medical Research and Development (AMED) (Grant 15gm0610007h0003).

## Author contributions

HH, ME, and YO designed the experiments. HH, ME, TK, and JM performed the experiments and/or provided advice and technical expertise. HH and KK carried out organoid culture analysis. KT provided unique reagents. KA and KM provided *Angptl2*$^{-/-}$ mice. HH, ME, and YO wrote the manuscript, and all authors amended the manuscript.

## Conflict of interest

The authors declare that they have no conflict of interest.

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
