## [Review Process File · The EMBO Journal]

Manuscript EMBO-2016-95690

ANGPTL2 expression in the intestinal stem cell niche controls epithelial regeneration and homeostasis

Haruki Horiguchi, Motoyoshi Endo, Kohki Kawane, Tsuyoshi Kadomatsu, Kazutoyo Terada, Jun Morinaga, Kimi Araki, Keishi Miyata and Yuichi Oike

Corresponding author: Haruki Horiguchi, Motoyoshi Endo, and Yuichi Oike, Kumamoto University

Review timeline:

Submission date:	09 September 2016
Editorial Decision:	18 October 2016
Revision received:	24 November 2016
Editorial Decision:	29 November 2016
Revision received:	30 November 2016
Accepted:	30 November 2016

Editor: Ieva Gailite

Transaction Report:

1st Editorial Decision

18 October 2016

Thank you for submitting your manuscript for consideration by the EMBO Journal. It has now been seen by two referees, whose comments are shown below. As you can see from the comments, both referees appreciate the novelty and significance of the proposed role of ANGPTL2 in intestinal homeostasis. Given the referees' very positive recommendations, I would like to invite you to submit a revised version of the manuscript, addressing the comments of the first referee.

When preparing your letter of response to the referees' comments, please bear in mind that this will form part of the Review Process File, and will therefore be available online to the community. For more details on our Transparent Editorial Process, please visit our website: http://emboj.embopress.org/about#Transparent_Process

Please let me know if you would like to discuss the revision at any point. Thank you for the opportunity to consider your work for publication. I look forward to your revision.

REFEREE REPORTS

Referee #1:

Within the manuscript 'ANGPTL2 expression in intestinal stem cell niche controls epithelial regeneration and homeostasis' the authors aim to study the role of ANGPTL2 in response to intestinal damage. To study this response, they have implemented two different well-established *in vivo* damage models; DSS-induced damage and radiation damage. They compare the regeneration response in wildtype versus ANGPTL2^{-/-} mice. Both models show that regeneration is impaired in the ANGPTL2 null mice, whereas in the absence of damage no significant differences could be observed apart from reduced expression of ISC markers/Wnt target genes, indicating a function for ANGPTL2 during regeneration rather than in normal tissue homeostasis. Subsequently they localized the expression of ANGPTL2 to the intestinal mesenchyme, especially to the intestinal subepithelial myofibroblasts (ISEMFs). The authors uncovered the pathways that are activated within ISEMFs in the presence and absence of ANGPTL2, however unfortunately not the direct relation between ISEMFs and the depletion of ISCs within the niche. The manuscript is clearly written and the data well presented. The experiments appear to be well conducted.

Several aspects that might improve the manuscript:

- Is ANGPTL2 upregulated in response to damage (DSS and radiation) in WT mice.?
- The organoid cultures they used in figure 7 do not look very viable. Do these represent a regenerative system or homeostasis? This needs to be discussed.
- In addition to the above, it would be relevant to see how these organoids would behave in response to e.g. irradiation. Furthermore, the authors should attempt to modify the media composition and coculture organoids of various genotype with fibroblasts to investigate if the *in vivo* findings can be recapitulated *in vitro* to allow for future complete disentanglement of the signals involved.
- The discussion on the role of BMP signaling needs to be expanded and the most recent literature incorporated.

Referee #2:

Horiguchi and colleagues study the function of angiopoietin-like protein 2 in the intestine of mice. The authors show that loss of ANGPTLII results in no significant phenotypes under homeostatic conditions, but that there is a decline in stem cell function and proliferation when ANGPTLII mutants are exposed to DSS or gamma irradiation. The authors show that ANGPTLII is expressed in intestinal subepithelial myofibroblasts and that these cells, and not the hematopoietic system, contribute to the niche of intestinal stem cells. These cells over-express BMP2 and BMP7 in ANGPTLII loss of function conditions and the authors propose that this induction of BMPs and the activation of BMP signaling in the epithelium limits ISC activity during regeneration. The authors further show that inhibition of integrin signaling can induce BMP2 and BMP7 in ISEMFs. Since previous data suggest that ANGPTLII binds to integrins, and since a direct effect on Wnt signaling is ruled out here, the authors propose a model in which Integrin/NFκB signaling in ISEMFs regulates BMP expression to influence the ISC niche and promote ISC proliferation in intestinal crypts.

These findings are of broad interest as they describe a new signaling interaction by which the intestinal stem cell niche and ISC quiescence are regulated through an interaction between ISEMFs and the epithelium under stress conditions.

The data presented are of high quality, the experiments are well-designed, and the conclusions are appropriate. Publication can be recommended as is.

Referee #1:

Within the manuscript 'ANGPTL2 expression in intestinal stem cell niche controls epithelial regeneration and homeostasis' the authors aim to study the role of ANGPTL2 in response to intestinal damage. To study this response, they have implemented two different well-established in vivo damage models; DSS-induced damage and radiation damage. They compare the regeneration response in wildtype versus ANGPTL2^{-/-} mice. Both models show that regeneration is impaired in the ANGPTL2 null mice, whereas in the absence of damage no significant differences could be observed apart from reduced expression of ISC markers/Wnt target genes, indicating a function for ANGPTL2 during regeneration rather than in normal tissue homeostasis. Subsequently they localized the expression of ANGPTL2 to the intestinal mesenchyme, especially to the intestinal subepithelial myofibroblasts (ISEMFs). The authors uncovered the pathways that are activated within ISEMFs in the presence and absence of ANGPTL2, however unfortunately not the direct relation between ISEMFs and the depletion of ISCs within the niche. The manuscript is clearly written and the data well presented. The experiments appear to be well conducted. Several aspects that might improve the manuscript:

We thank you for your comments, which were very helpful. We have extensively revised the manuscript and addressed your comments. In the revision, we added data related to co-culture of organoids with ISEMFs of respective genotypes and show a direct relationship between ISEMFs and organoid formation in vitro. We feel that these results strengthen our conclusions.

1-Is ANGPTL2 upregulated in response to damage (DSS and radiation) in WT mice.?

Yes. We found that ANGPTL2 expression in WT mice was upregulated in response to DSS- or irradiation-induced damage (Fig EV2A and EV3A). We report these findings on page 10, lines 2-4 and page 11, lines 8-10 of the revised manuscript.

2-The organoid cultures they used in figure 7 do not look very viable. Do these represent a regenerative system or homeostasis? This needs to be discussed.;

As you note, organoids shown in Figure 7A (right panel) and Figure 7C in the original paper appear damaged. We conclude that damage is attributable to either Wnt deficiency or in some cases due to passage through a syringe. By contrast, a representative normal organoids shown in Figure 7A (left panel) resembles one reported by others (Mahe et al., 2013; Sato et al., 2011; Yui et al., 2012). Thus we feel that the organoid culture used in this paper was a valid system in which to test our hypotheses.

As previously reported (Chiacchiera et al., 2016a; Date & Sato, 2015; Oudhoff et al., 2016), organoid culture systems mimic conditions of tissue regeneration. To examine a potential ANGPTL2 function in intestinal regeneration, we employed this model. In the revision, we now describe organoid culture systems and state explicitly in the Discussion (page 24, lines 3-10) that "Organoid formation from isolated crypts reportedly mimics conditions of tissue regeneration (Chiacchiera et al., 2016b; Date & Sato, 2015; Oudhoff et al., 2016). Accordingly, we feel that our comparable analysis of colonic organoid formation in the presence or absence of recombinant ANGPTL2 protein serves as a useful system in which to test a potential paracrine role for ANGPTL2 in regulating intestinal regeneration."

3-In addition to the above, it would be relevant to see how these organoids would behave in response to e.g. irradiation. Furthermore, the authors should attempt to modify the media composition and coculture organoids of various genotype with fibroblasts to investigate if the in vivo findings can be recapitulated in vitro to allow for future complete disentanglement of the signals involved.;

Thank you for these comments. Based on your suggestion, we have added data testing organoid response to irradiation (Fig EV6E and EV6F). Although the number of viable organoids decreased after irradiation, we observed no difference in crypt formation capacity with or without rANGPTL2 treatment (Figs EV6C-6F) effects similar to those seen after mechanical damage of organoids

caused by passage through a syringe. These observations suggest that ANGPTL2 derived from ISEMFs does not directly alter the IEC regenerative response. These findings are reported on page 17, lines 6-12 of the revised manuscript.

To examine ISEMF function in organoid culture, we co-cultured wild-type or *Angptl2*^{-/-} mouse crypts with ISEMFs derived from mice of either genotype using two paradigms: either direct contact (Fig 7C) or separated by a transwell membrane (Fig EV7A). We observed no difference in either growth or size of organoids in the presence of wild-type or *Angptl2*^{-/-} ISEMFs in either paradigm under optimized culture conditions, which included exogenous Noggin (Nog (+)) (Figs 7D, 7E, EV7B and EV7C). However, we hypothesized that ISEMF-derived BMP might be antagonized by exogenous and ISEMF-derived Noggin. Thus, we conducted the same experiments without exogenous Noggin. We found that the size of organoids of either genotype co-cultured with *Angptl2*^{-/-} ISEMFs decreased compared with wild-type ISEMFs in either paradigm under culture conditions lacking exogenous Noggin (Nog (-)) (Figs 7D, 7E Figs EV7B and EV7C). These findings suggest that abundant BMP secreted from *Angptl2*^{-/-} ISEMFs might inhibit organoid growth. These findings are reported on page 18, line 3-page 19, line 3 of the revised manuscript.

We also treated organoids with ISEMF conditioned medium (CM) (Fig EV7D). Crypt size of organoids treated with *Angptl2*^{-/-} ISEMF CM was minimally decreased relative to tissues treated with wild-type ISEMF CM (Figs EV7E and EV7F), suggesting that secretion of soluble BMP from *Angptl2*^{-/-} ISEMFs may decrease organoid growth.

These results suggest that impaired regeneration seen in *Angptl2*^{-/-} mouse crypts is likely due to upregulated expression of ISEMF-derived BMP (Fig 7F). These findings are reported on page 19, lines 3-7 of the revised manuscript.

4-The discussion on the role of BMP signaling needs to be expanded and the most recent literature incorporated.;

Thank you for this suggestion. We have now added description relevant to BMP signaling to the Discussion, incorporating the most recent literature. These findings are reported on page 22, lines 2-5 and page 22, lines 8-page 23, line 6 and page 23, lines 18-page 24, line 2 of the revised manuscript.

Referee #2:

Horiguchi and colleagues study the function of angiopoietin-like protein 2 in the intestine of mice. The authors show that loss of ANGPTLII results in no significant phenotypes under homeostatic conditions, but that there is a decline in stem cell function and proliferation when ANGPTLII mutants are exposed to DSS or gamma irradiation. The authors show that ANGPTLII is expressed in intestinal subepithelial myofibroblasts and that these cells, and not the hematopoietic system, contribute to the niche of intestinal stem cells. These cells over-express BMP2 and BMP7 in ANGPTLII loss of function conditions and the authors propose that this induction of BMPs and the activation of BMP signaling in the epithelium limits ISC activity during regeneration. The authors further show that inhibition of integrin signaling can induce BMP2 and BMP7 in ISEMFs. Since previous data suggest that ANGPTLII binds to integrins, and since a direct effect on Wnt signaling is ruled out here, the authors propose a model in which Integrin/NFkB signaling in ISEMFs regulates BMP expression to influence the ISC niche and promote ISC proliferation in intestinal cypts.

These findings are of broad interest as they describe a new signaling interaction by which the intestinal stem cell niche and ISC quiescence are regulated through an interaction between ISEMFs and the epithelium under stress conditions.

The data presented are of high quality, the experiments are well-designed, and the conclusions are appropriate. Publication can be recommended as is.

Thank you very much for your evaluation of our manuscript.

Cited references

- Chiacchiera F, Rossi A, Jammula S, Zanotti M, Pasini D (2016b) PRC2 preserves intestinal progenitors and restricts secretory lineage commitment. *EMBO J* 35: 2301-2314
- Date S, Sato T (2015) Mini-gut organoids: reconstitution of the stem cell niche. *Annu Rev Cell Dev Biol* 31: 269-289
- Mahe MM, Aihara E, Schumacher MA, Zavros Y, Montrose MH, Helmrath MA, Sato T, Shroyer NF (2013) Establishment of Gastrointestinal Epithelial Organoids. *Curr Protoc Mouse Biol* 3: 217-240
- Oudhoff MJ, Braam MJ, Freeman SA, Wong D, Rattray DG, Wang J, Antignano F, Snyder K, Refaeli I, Hughes MR, McNagny KM, Gold MR, Arrowsmith CH, Sato T, Rossi FM, Tatlock JH, Owen DR, Brown PJ, Zaph C (2016) SETD7 Controls Intestinal Regeneration and Tumorigenesis by Regulating Wnt/beta-Catenin and Hippo/YAP Signaling. *Dev Cell* 37: 47-57
- Sato T, Stange DE, Ferrante M, Vries RG, Van Es JH, Van den Brink S, Van Houdt WJ, Pronk A, Van Gorp J, Siersema PD, Clevers H (2011) Long-term expansion of epithelial organoids from human colon, adenoma, adenocarcinoma, and Barrett's epithelium. *Gastroenterology* 141: 1762-1772
- Yui S, Nakamura T, Sato T, Nemoto Y, Mizutani T, Zheng X, Ichinose S, Nagaishi T, Okamoto R, Tsuchiya K, Clevers H, Watanabe M (2012) Functional engraftment of colon epithelium expanded in vitro from a single adult Lgr5(+) stem cell. *Nat Med* 18: 618-623

2nd Editorial Decision

29 November 2016

Thank you for submitting a revised version of your manuscript. The manuscript has now been seen by Referee #1, who finds that all his/her concerns and recommendations have been addressed. Therefore I am happy to accept the manuscript in principle. However, before I can officially send you the acceptance letter, there are a few editorial issues concerning the text and figures that I need you to address in a final revision.

Thank you again for giving us the chance to consider your manuscript for The EMBO Journal. I look forward to your final revision.

2nd Revision - authors' response

30 November 2016

Authors made requested editorial changes.

Corresponding Author Name: Motoyoshi Endo

Journal Submitted to: The EMBO Journal

Manuscript Number: EMBOJ-2016-95690